# Recombinant Influenza A Viruses Expressing Reporter Genes from the Viral NS Segment

**DOI:** 10.3390/ijms251910584

**Published:** 2024-10-01

**Authors:** Luis Martinez-Sobrido, Aitor Nogales

**Affiliations:** 1Texas Biomedical Research Institute, San Antonio, TX 78227, USA; 2Center for Animal Health Research, CISA-INIA-CSIC, 28130 Madrid, Spain

**Keywords:** recombinant influenza A virus, plasmid-based reverse genetics, reporter genes, fluorescence, luminescence, replication-competent reporter-expressing influenza A virus, NS segment

## Abstract

Studying influenza A viruses (IAVs) requires secondary experimental procedures to detect the presence of the virus in infected cells or animals. The ability to generate recombinant (r)IAV using reverse genetics techniques has allowed investigators to generate viruses expressing foreign genes, including fluorescent and luciferase proteins. These rIAVs expressing reporter genes have allowed for easily tracking viral infections in cultured cells and animal models of infection without the need for secondary approaches, representing an excellent option to study different aspects in the biology of IAV where expression of reporter genes can be used as a readout of viral replication and spread. Likewise, these reporter-expressing rIAVs provide an excellent opportunity for the rapid identification and characterization of prophylactic and/or therapeutic approaches. To date, rIAV expressing reporter genes from different viral segments have been described in the literature. Among those, rIAV expressing reporter genes from the viral NS segment have been shown to represent an excellent option to track IAV infection in vitro and in vivo, eliminating the need for secondary approaches to identify the presence of the virus. Here, we summarize the status on rIAV expressing traceable reporter genes from the viral NS segment and their applications for in vitro and in vivo influenza research.

## 1. Introduction

### 1.1. Influenza Viruses

Influenza viruses are classified into influenza A (IAV), B (IBV), C (ICV), and D (IDV) subtypes within the *Orthomyxoviridae* family [1]. To date, only IAV and IBV are responsible for seasonal epidemic, and only IAVs have been responsible for occasional pandemics of great consequence to humans [2,3,4]. Usually, IAV pandemics occur when a new IAV subtype is introduced into the human population due to the ability of IAV to infect multiple avian and mammalian species [2]. Both prophylactics and therapeutics measures are available to prevent or treat, respectively, IAV infections [1,3,5,6,7]. However, IAV can escape pre-existing immunity induced by prophylactic vaccines and became resistant to therapeutic antivirals due to its ability to rapidly evolve [1,3]. Thus, there is an urgent need to develop new vaccines and to identify novel antivirals to efficiently combat IAV infections, a process that will be facilitated by the use of recombinant (r)IAV expressing reporter genes [7,8,9,10,11,12].

IAVs are negative-sense, single-stranded RNA viruses of negative polarity (Figure 1). IAV virion is made of the two major viral glycoproteins, hemagglutinin (HA), encoded by segment 4; and neuraminidase (NA), encoded by segment 6, responsible for viral entry and release from infected cells, respectively. Also, in the virion is located the ion channel matrix 2 (M2) protein, encoded by the M segment 7, that is important in the initial steps of viral replication. Under the viral envelope is the matrix 1 (M1) protein, also encoded by the M segment 7, that is essential for viral budding. Inside the viral particle locates the 8-genome viral (v)RNA segments (PB2, PB1, PA, HA, NP, NA, M, and NS) that are encapsidated by the viral nucleoprotein (NP), encoded by segment 5, and associated to one copy of the viral polymerase subunits that form the viral ribonucleoprotein (vRNP) complexes. The viral RNA-dependent RNA polymerase (RdRp) complex is formed by three polymerase subunits: PB2, encoded by segment 1; PB1, encoded by segment 2; and PA, encoded by segment 3, and are involved in viral genome replication and gene transcription [13,14,15]. Under the viral envelope is also the nuclear export protein (NEP), encoded by NS segment 8, that is required for the nuclear export of vRNP complexes from the nucleus to the cytoplasm of infected cells. Flanking the coding regions in each of the vRNA segments are the 3′ and 5′ non-coding regions (NCRs) important for viral replication and transcription [14,15]. In addition, the 3′ and 5′ ends of the coding regions contain the packaging signals that are required, together with the NCR, for the encapsidation of the viral genome into nascent viruses [13,14,15].

### 1.2. IAV Life Cycle

IAV life cycle can be divided into viral attachment to the sialic acid receptor, entry of the virus into the cell using an endocytosis-mediated mechanism, fusion of the virus membrane with the membrane of the endosome, nuclear export of vRNAs from the cytoplasm to the cell nucleus, viral replication and transcription, nuclear export of vRNAs from the nucleus to the cell cytoplasm, packaging of vRNAs into nascent virions, budding of new viral particles, and virus release from infected cells (Figure 2) [16,17]. The first step in the IAV life cycle starts with the attachment of the viral particle to sialic acid cellular receptors present in susceptible cells, a process mediated by the viral HA [18]. In general, avian IAV recognizes α-(2,3), while mammalian AIV recognizes α- (2,6) sialic acid receptors containing glycoproteins. The ability of swine IAV to recognize both α-(2,3) and α-(2,6) sialic acid receptors is the reason why swine species represent a mixing vessel for avian and mammalian IAVs and the concern for the origin of new IAVs with pandemic potential [19,20,21]. After attachment to the sialic acid-containing receptor, IAV enters the cell using a receptor-mediated endocytosis mechanism [22]. Then, acidification of the pH of the endosome, a process mediated by the viral M2 protein, triggers a conformational change in the viral HA that allows the fusion of the membrane of the virus with the membrane of the endosome [23,24]. This fusion event results in the release of vRNPs into the cell cytoplasm [23]. Once in the cytoplasm, the vRNPs are translocated to the nucleus where viral replication and transcription take place [25]. Nuclear import of vRNPs is a process mediated by the interaction of the nuclear localization signal (NLS) of IAV proteins that make the vRNP complexes (PB2, PB1, PA, and NP) with karyopherin-like importin α and β [25]. Inside the nucleus, the negative-sense vRNA is first transcribed into a positive-sense mRNA that is translated into viral proteins. The same negative-sense vRNA serves as a template to provide a complimentary copy of positive-sense RNA (cRNA) that is used to generate new vRNAs that will be incorporated into nascent virions [15,25]. Viral genome replication and gene transcription is mediated by the viral polymerase subunits PB2, PB1, and PA, together with the viral NP [15,25]. Newly generated vRNAs are next exported to the cell cytoplasm using a Crm1-dependent pathway, a process mainly mediated by interaction of the viral NEP with Crm1 and M1 [26,27]. Once in the cytoplasm, the vRNAs are selectively packaged into nascent virions [28]. Specific packaging of newly synthesized vRNAs into IAV is a process mediated by the packaging signals located at the 3′ and 5′ NCRs of each of the viral segments. This specific packaging mechanism guarantees the incorporation of one copy of each of the vRNAs into new virions [29,30,31]. Next, IAV particles leave the infected cells from the apical side using a budding mechanism [32,33]. Budding of newly synthesized virions from infected cells is mainly mediated by the interaction of the M1 protein with newly synthesized vRNPs and viral glycoproteins in the membrane of infected cells [34,35]. Cleavage of sialic acid residues from newly synthesized HA present in budding virions is required for the release of IAV from infected cells, a process mediated by the viral NA [36,37].

### 1.3. IAV NS Segment

IAV segment 8 encodes the NS mRNA as a continuous primary transcript, and its regular processing produces the NS1 protein, which is highly expressed at the beginning of viral infection. On the other hand, alternative processing using a weak 5′ alternative splice site results in a less abundant splice gene encoding the viral NEP. Although both proteins are translated from different open reading frames (ORFs), they still share the first 10 N-terminal amino acids, including the initiation codon (AUG) [1,15,38].

IAV NS1 is a multifunctional protein and virulence factor that is generally divided into at least two distinct alleles [6,39]. NS1 allele “A” is most often found in all IAV isolates from mammals and some avian strains, whereas NS1 allele “B” is restricted to some avian IAVs [40]. IAV NS1 contributes to viral replication and pathogenicity by suppressing interferon (IFN) production and the antiviral activity of many IFN-stimulating genes (ISGs) [6,41,42,43,44,45]. IAV NS1 also contributes to the shutdown of host gene expression [40,42]. IAV NS1 protein interacts with a high number of cellular proteins as well as some viral proteins to modulate viral replication and innate immune responses [39]. Importantly, NS1 of different IAV subtypes can have different properties or functions and shows differential interaction patterns with cellular host factors [46,47].

IAV NEP [48] is found in association with the M1 protein within the virion and is involved in mediating the nuclear export of viral vRNP complexes in the late stage of the viral replication cycle [1]. In addition, NEP contributes to the viral budding process [49], and it can control the accumulation of vRNA species leading to transition from early transcription to genome replication and vRNP production [50].

### 1.4. Reverse Genetics of IAV

IAV reverse genetics refers to the ability to generate rIAV from plasmid DNA [5,6,51]. These reverse genetics techniques have allowed the generation of recombinant viruses to assess different aspects in the biology of IAV, including the function of viral proteins and the contribution of specific amino acids in protein functions, virus–host interactions, viral replication and pathogenesis, among others [42,43,46]. Likewise, reverse genetics techniques have allowed to identify nucleotide domains or residues required for viral packaging, viral genome replication, and gene transcription [52,53,54]. Furthermore, reverse genetics has been used to develop inactivated or live-attenuated influenza vaccines [5,6,12,51,53]. Reverse genetics has also allowed the generation of single-cycle infectious rIAV to safely study highly pathogenic viruses under less restricted biosafety level (BSL) laboratory conditions [12,53]. Notably, reverse genetics has been used to generate rIAVs expressing foreign genes, including reporter fluorescent and luciferase proteins, to easily identify the presence of the virus in cultured cells and validated animal models of infection [9,55,56,57,58]. In this chapter, we will focus on the strategies to generate rIAV expressing reporter genes from the viral NS or segment 8.

IAV reverse genetics requires the expression of the eight negative-stranded vRNAs and the viral components required for viral genome replication and gene transcription (PB2, PB1, PA, and NP) from the same plasmid DNA using an ambisense or bidirectional approach. The most common IAV reverse genetics is based on the transfection of eight ambisense plasmids, one for each of the IAV segments, built on the use of polymerase I (Pol I) and polymerase II (Pol II) cassettes to express vRNAs and proteins, respectively, from transfected cells [5,51,53,59,60]. The Pol I cassette is made of the Pol I promoter and the Pol I terminator and is flanked by the Pol II cassette that is made of a Pol II promoter and polyademinalion signal [5,51,53,59,60]. While the Pol II promoter can be used in different cell lines, the Pol I promoter is species-specific. Currently, the most common IAV reverse genetics is based on the use of a human Pol I promoter and human cells. By transfecting the eight ambisense plasmids into co-culture of human 293T cells, known to have a high transfection efficiency, and MDCK cells, known to be the best substrate to amplify IAV in cell culture (Figure 3) [5,51,53,59,60], rIAV can be successfully generated and used to study multiple aspects of the biology of IAV, including the generation of reporter-expressing rIAV.

## 2. rIAV Expressing Reporter Genes from the NS Segment

### 2.1. rIAV Expressing Reporter Genes from a Modified NS Segment

rIAV expressing reporter genes from different vRNA segments have been described in the literature, including fluorescent and luciferase reporter genes [53,55,62,63,64,65,66,67,68,69,70]. Among those, rIAV expressing reporter genes from the NS segment have been shown to represent an excellent option to track IAV infection in vitro and in vivo, eliminating the need for secondary approaches to identify the presence of the virus (Table 1) [9,55,56,71,72,73]. Some of the advantages of expressing reporter genes from the viral NS segment include its small size (~890 nt) and therefore its potential ability to tolerate bigger insertions compared to other vRNA segments; the ability to express reporter genes without the need for disrupting and duplicating packaging signals located at the 3′ or 5′ end of each vRNA for efficient virion assembly; its ability to maintain the inserted reporter gene since other replication-competent rIAVs have been shown to easily lose reporter gene expression after viral passaging; and the high levels of NS1 expression during viral infection. Because of all these advantages, the NS segment of multiple IAV strains has been used for developing reporter-expressing rIAV due to the deep knowledge accumulated about the expression strategy of the NS segment and the functions of its gene products (Table 1). Importantly, recombinant constructs need to maintain intact the NCRs of viral genes and the packaging signals to recover infectious viruses. Figure 4 shows one of the strategies used to modify the NS segment to generate reporter-expressing rIAV. In this strategy, the NS segment is modified to encode NS1 and NEP from a single non-overlapping transcript. To that end, the overlapping region of the viral genome encoding the N-terminal 10 amino acids shared between NS1 and NEP is duplicated, and the porcine teschovirus 1 (PTV-1) 2A autoproteolytic cleavage site is inserted between the NS1 and NEP ORFs. This strategy ensured that NS1 and NEP are translated as two separate, non-overlapping, independent ORFs. However, the IAV NS segment has also been modified using other strategies to encode reporter genes or foreign sequences, including the insertion of two 2A autoproteolytic cleavage sequences for the individual expression of viral proteins and the reporter gene [44,45,57,73]. Likewise, expression of foreign reporter genes has been accomplished using caspase recognition sites and stop/start sequences [74,75]. Notably, there have not been reported differences in the life cycle of natural isolates of IAV, or wild-type rIAV, and reporter-expressing rIAV.

The feasibility to generate rIAV expressing reporter genes has been a significant advancement in studying IAV. These include the feasibility to track viral infections in cultured cells without the need for secondary approaches to identify infected cells [9,55,56,71,72,73]. Likewise, rIAV expressing reporter genes are compatible with high-throughput screening (HTS) approaches for the identification of antivirals or neutralizing antibodies [7,11,38,61], which could help to identify new therapeutic options against IAV or to evaluate the efficacy of novel vaccine prototypes. Moreover, rIAV expressing reporter genes can be used to easily track viral infections in vivo using in vivo imaging systems (IVISs) [11,56,71,72,73,76,77]. In addition, IVISs can be used to identify the presence of the virus ex vivo from tissues from infected mice [11,38,71,72]. These in vivo and ex vivo applications of rIAV expressing reporter genes have revolutionized the field of influenza research. Importantly, the IAV strain and the reporter gene selected to generate the reporter-expressing rIAV can affect the in vitro and/or in vivo properties of the virus, including the limit of detection, pathogenicity, and virus propagation, among others.

The two major classes of reporter genes used to generate replication-competent rIAV include fluorescent and luciferases proteins. Both reporter genes have advantages and disadvantages [82,83,84]. For instance, fluorescent proteins are likely a better option for in vitro studies aimed at detecting the presence of the virus in infected cells. However, for in vitro quantitative purposes, luciferase proteins represent a better option. For in vivo studies, luciferase proteins represent a better option to detect the presence of the virus in the entire infected animals using IVISs. However, they required the administration of luciferase substrates, which are usually expensive and require additional manipulation of the animals for injection of the substrate. Thus, for ex vivo imaging, fluorescent proteins may represent a better option to luciferase proteins since fluorescent expression can be easily detected ex vivo using IVISs without the need for administering a luciferase substrate, although animals need to be sacrificed to collect organs and/or tissues.

### 2.2. rIAV Expressing Reporter Genes from Modified NS and HA Segments

Previous studies have been limited to the expression of a reporter fluorescent or luciferase gene from a viral segment, including NS [9,10,11,28,53,55,56,63,66,67,71,72,73,76,77,83,85,86,87,88]. This has forced researchers to select the use of fluorescent or luciferase proteins to generate rIAV, limiting the capability of using two reporter genes to overcome the obstacles of using a single reporter rIAV. However, this obstacle has been overcome with the generation of replication-competent rIAV expressing two reporter genes from two different viral segments [7,11,72]. Nogales et al. generated an rIAV expressing two reporter genes, a fluorescent protein from the NS segment and a nanoluciferase (Nluc) protein from the HA segment (Figure 5). This bireporter rIAV (BIRFLU) was able to express both reporter genes with similar growth kinetics to wild-type virus in cultured cells. However, in vivo, BIRFLU was attenuated as compared to wild-type IAV, but the authors were able to detect Nluc expression in vivo using IVISs and Venus expression ex vivo in the lungs of infected mice [11,72]. Notably, the authors were able to demonstrate the feasibility of using this rIAV expressing reporter fluorescent and luciferase proteins to easily track viral infection in cultured cells and for the identification and characterization of antivirals and neutralizing antibodies [7,11,72]. Importantly, BIRFLU was genetic and phenotypic stable up to 4 passages in cultured cells, and expression of fluorescent proteins from the NS segment was not limited to Venus since the authors were able to also generate a BIRFLU expressing mCherry in addition to Nluc from the HA segment [11]. The ability to generate rIAV expressing two reporter genes overcomes the need for selecting a single reporter to generate recombinant reporter-expressing IAV. More importantly, this BIRFLU IAV takes the advantages of both fluorescent and luciferase reporter genes improving basic and translational research, in addition to demonstrating extra plasticity in the IAV genome to express two foreign genes. BIRFLU can represent an excellent option in HTS to identify antivirals to avoid compounds that interfere with one of the reporter genes that could result in the identification of false-positive hits [7]. Moreover, the feasibility of expressing two foreign genes from the IAV genome opens the possibility to develop better IAV vaccines or the feasibility of using IAV as a vaccine vector for the treatment of other viral infections.

### 2.3. rIAV Expressing Reporter Genes Instead of the Viral NS1 Protein

rIAV expressing reporter genes fused to the viral NS1 protein has allowed to circumvent the requirement of secondary approaches to identify the presence of the virus in infected cells [9,11,53,55,56,57,73,76,77,88]. Contrary to wild-type viruses, IAVs lacking the NS1 protein are attenuated in viral replication in most cells, since they lack the ability to modulate IFN and ISG responses [6,40,42,86,89]. Recently, Nogales et al. generated a replication-competent NS1-deficient rIAV expressing fluorescent or Nluc reporter genes from the NS segment by substituting the NS1 ORF with that of the reporter gene (Figure 6) [86]. As expected, the NS1-deficient rIAV was viable and easy to track using fluorescent microscopy (fluorescent) or a luciferase plate reader (Nluc) but was attenuated in viral replication, as compared with wild-type IAV, in cultured cells and in vivo [86]. This was the first IAV expressing reporter genes from the locus of a modified NS segment lacking the NS1 protein that can be used, instead of the previously described ΔNS1 IAV lacking expression of reporter genes, for in vitro or in vivo studies. One of the limitations of using ΔNS1 IAV is its impairment in viral replication due to the lack of NS1 to control IFN production and/or signaling, and ISGs [39,42,89]. This ΔNS1 rIAV expressing reporter genes represents an excellent opportunity to evaluate viral replication in cultured cells and to study different aspects in the biology of IAV lacking NS1, including its ability to induce IFN responses. Moreover, the feasibility of expressing other foreign genes instead of the viral NS1 opens the possibility for implementing NS1-deficient rIAV as a vaccine or as a vaccine vector for the treatment of other viral infections.

## 3. Conclusions and Future Directions: Recombinant

Replication-competent rIAV expressing reporter genes from different viral segments have been described in the literature and used in both basic and translational research [9,52,55,56,58]. These reporter-expressing rIAVs have been used to study different aspects in the biology of IAV as well as to identify neutralizing antibodies, antivirals, or host factors required for viral entry, genome replication, or gene transcription, or budding [7,10,38,53,55,71]. Importantly, these reporter-expressing, replication-competent rIAVs have allowed for easily tracking viral infections in vitro and in vivo without the need for secondary approaches to identify the presence of the virus in infected cells or validated animal models, therefore significantly improving our knowledge of IAV replication and pathogenesis [38,56,72,76,88]. The presence of IAV in infected cells can be easily tracked using reporter virus expressing fluorescent proteins with in vitro imaging (e.g., fluorescent microscopy). Likewise, IAV can be easily quantified using luciferase-expressing viruses in conjunction with luciferase plate readers. In vivo, the presence of IAV in infected mice or other animal models can be detected by using IVISs. Ex vivo, IAV can be detected in the lungs of infected animals by fluorescent or luciferase expression using IVISs. Importantly, the use of luciferase-expressing rIAV and IVISs for in vivo studies has allowed researchers to reduce the number of animals used in each experiment since viral infection can be monitored from the same animals without the need to euthanize them to evaluate viral replication at different times after infection. Therefore, the use of reporter-expressing rIAV also has important ethical benefits. Following the example of IAV, rIBV expressing reporter genes from the NS segment have also been described in the literature (Table 2) [71,90].

However, the use of replication-competent, reporter-expressing rIAV from the viral NS segment also has limitations. One of the limitations is that expression of reporter genes from the locus of the viral NS segment usually results in viral attenuation [38,56,72,76,88]. It has been shown that expression of luciferase reporter genes, mainly Nluc, results in less viral attenuation than expression of fluorescent proteins [86]. Another limitation is the stability of the reporter gene after serial passage of the reporter-expressing rIAV. This stability issue can be related to the reporter gene used and/or the IAV, or IBV, strain used. Therefore, it is important to monitor and confirm expression of reporter genes from reporter-expressing rIAV preparations to guarantee uniformly expressing the reporter genes from the viral preparation.

## Figures and Tables

**Figure 1 ijms-25-10584-f001:**
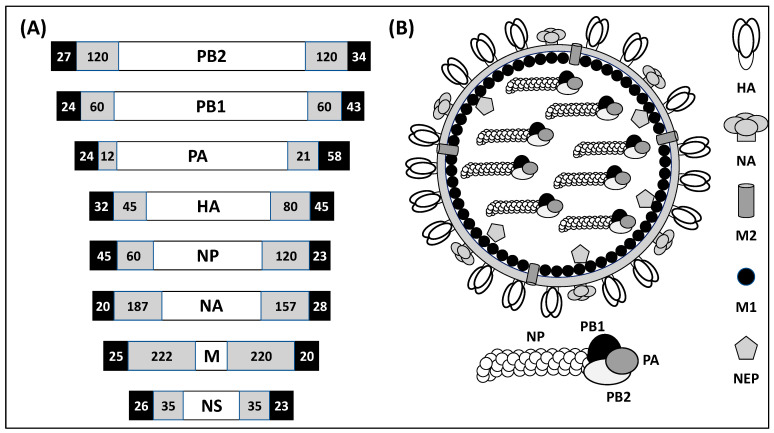
IAV genome organization and viral particle structure. (**A**) Genomic organization: The IAV genome is made of eight negative-sense, single-stranded vRNA segments. Each viral segment contains the 3′ and 5′ non-coding regions (NCRs) involved in viral genome replication and gene transcription (black boxes). Together with the NCR, the 3′ and 5′ ends of the vRNA coding regions (gray boxes) contain the packaging signals required for efficient incorporation of the vRNAs into the new virions. The genome of IAV is organized based on the length of the vRNAs as PB2, PB1, PA, HA, NP, NA, M, and NS. (**B**) Viral particle structure: The surface of the influenza viral particle is made of a lipid bilayer decorated with the three major surface proteins: HA, NA, and M2. Under the viral membrane is the M1 protein. Inside the viral particle are located the NEP and the vRNAs encapsidated by the viral NP. Each of the vRNAs contains the three components of the viral RdRp subunits (PB2, PB1, and PA).

**Figure 2 ijms-25-10584-f002:**
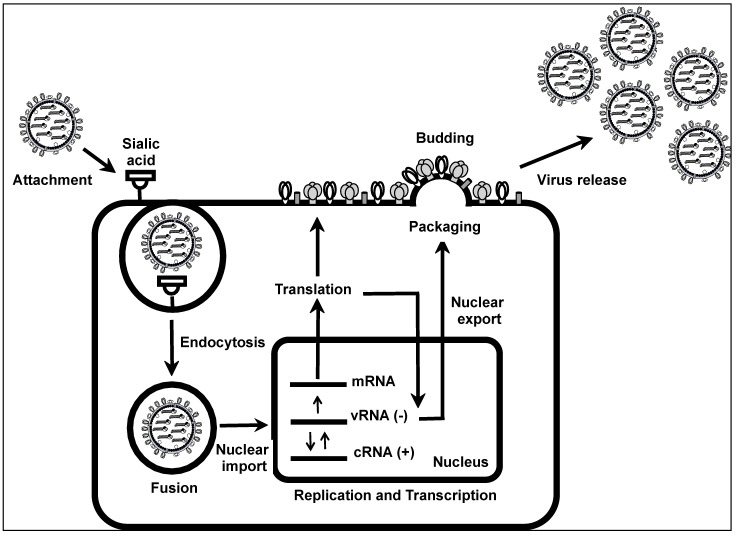
Life cycle of IAV: The different steps of the IAV life cycle include the attachment of the virus into susceptible cells, a process mediated by the interaction of IAV HA to sialic acid-containing receptors. Next, IAV is internalized into the cytoplasm of the cell using an endocytosis-mediated mechanism. Fusion of the viral membrane with the membrane of the endosome, a process mediated by the viral HA and M2, results in the release of vRNPs into the cytoplasm of infected cells. Nuclear import of vRNPs is mediated by the interaction of NLS present in the viral polymerase proteins and NP with importin α and β. Inside the nucleus, viral transcription and replication mediated by PB2, PB1, PA, and NP results in the formation of mRNAs that are translated in the cytoplasm of infected cells to produce viral proteins and the formation of complementary (c)RNAs that are used as a template to produce new vRNAs, respectively. Nuclear export of newly synthesized vRNAs is mediated by the interaction of NEP with Crm1 and M1. Inside the cytoplasm, vRNPs are selectively incorporated into nascent virions using specific packaging signals present in the 3′ and 5′ NCRs of each of the vRNAs. Budding of IAV is mediated by the interaction of M1 with newly synthesized vRNPs and viral glycoproteins in the membrane of infected cells. Finally, release of IAV from infected cells is mediated by cleavage of HA from sialic acid-containing glycoproteins in the cell membrane by NA.

**Figure 3 ijms-25-10584-f003:**
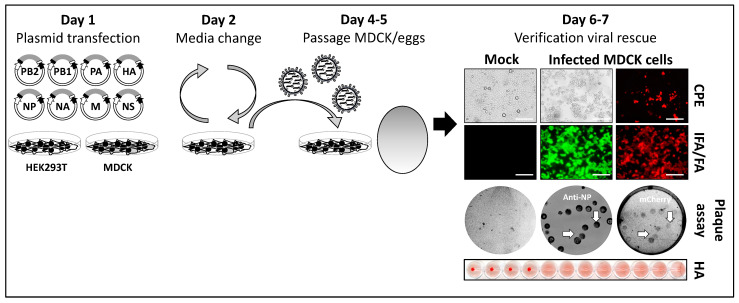
Plasmid-based reverse genetics to generate rIAV: Conventional reverse genetics to generate rIAV is based on the use of 8 plasmid DNAs. Each of the viral plasmids uses an ambisense strategy to encode the vRNAs and the viral proteins under Pol I and Pol II promoters, respectively. To rescue rIAVs, the 8 ambisense plasmids are co-transfected into co-cultures of 293T and MDCK cells (Day 1). After transfection, the cell culture supernatant is replaced by fresh media containing TPCK-trypsin (Day 2). At 2–3 days after transfection, the cell culture supernatant is collected and used to infect new MDCK cells or 8–10 chicken embryonated eggs (Day 4–5). Then, 2–3 days after infection, cell culture supernatant from infected MDCK cells or the allantoic fluid from chicken embryonated eggs is harvested to confirm the presence of rIAV. Detection of rIAV can be determined by evaluating cytopathic effect (CPE); immunofluorescence (IFA), using IAV-specific antibodies, or fluorescence assay (FA) if fluorescent-expressing viruses are rescued. Plaque assay with/without immunostaining with IAV-specific antibodies and fluorescence analysis is used to confirm expression of the reporter gene from all the recombinant viruses in the preparation (arrows indicate correlation between IAV viral NP and mCherry expression). Alternatively, a conventional hemagglutination assay (HA) can be used to demonstrate the presence of rIAV in the cell culture supernatants or allantoic fluid from MDCK and eggs, respectively [8,61]. IFA/FA scale bars 200 µm.

**Figure 4 ijms-25-10584-f004:**
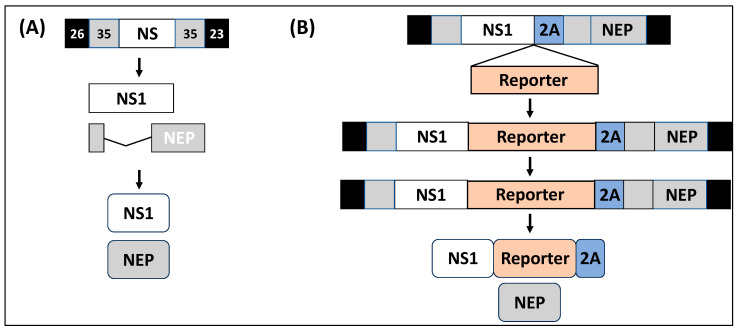
Modified IAV NS segment to express reporter genes. (**A**) IAV NS segment: IAV NS segment encodes the viral NS1 (white box) and NEP (gray box) using an alternative splicing mechanism. At the 3′ and 5′ ends of the NS segment are the NCRs (black boxes) involved in viral genome replication and gene transcription. Inside the coding region are the 3′ and 5′ packaging signals (gray boxes) required, together with the NCRs, for incorporation of the NS segment into new virions. (**B**) Modified IAV NS segment: To generate rIAV expressing reporter genes, the NS segment is modified to encode the viral NS1 (white box) and NEP (gray box) from a single transcript separated by the porcine teschovirus 1 (PTV-1) 2A autoproteolytic cleavage site (blue box). The first 10 amino acids shared between NS1 and NEP are duplicated downstream the 2A site (gray box). The foreign reporter gene (orange box) is cloned fused to the C-terminal of the NS1 protein and in frame with the viral NEP. The 2A peptide remains linked to the C-terminal of the NS1–reporter gene fusion.

**Figure 5 ijms-25-10584-f005:**
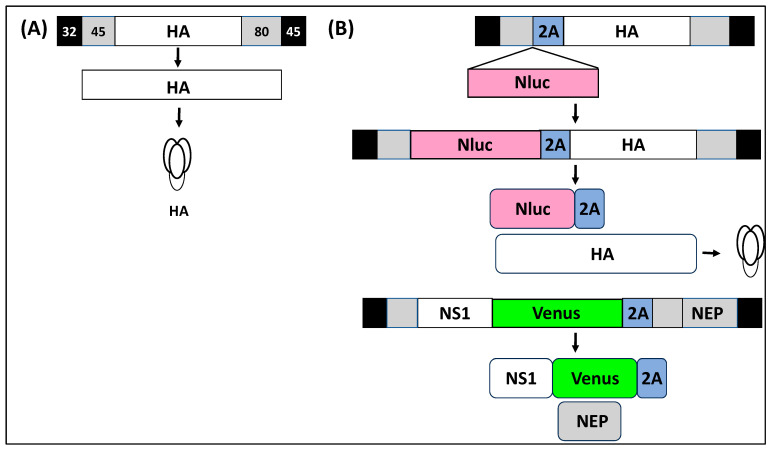
rIAV with modified NS and HA segments to express reporter fluorescent and luciferase genes. (**A**) IAV HA segment: IAV segment 4 encodes the viral HA (white box). At the 3′ and 5′ ends of the NS segment are the NCRs (black boxes) involved in viral genome replication and gene transcription. Inside the coding region are the 3′ and 5′ packaging signals (gray boxes) required, together with the NCRs, for efficient incorporation of the HA segment into nascent virions. (**B**) Modified IAV HA segment: In the case of the modified IAV HA segment, Nluc (pink box) is inserted as a fusion to the viral HA separated by the PTV-1 2A autoproteolytic cleavage site (blue box). The modified IAV NS segment to generate the BIRFLU IAV is identical to the one described in Figure 4. HA and NS modified segments are transcribed as single transcripts and translated as a single polyprotein. NS1 fused to Venus is separated from the NEP after cleavage in the PTV-1 2A site. Likewise, Nluc is separated from the viral HA after cleavage in the PTV-1 2A site. The 2A peptide remains fused at the C-terminal of the NS1–Venus fusion (green box, NS segment) and Nluc (HA segment).

**Figure 6 ijms-25-10584-f006:**
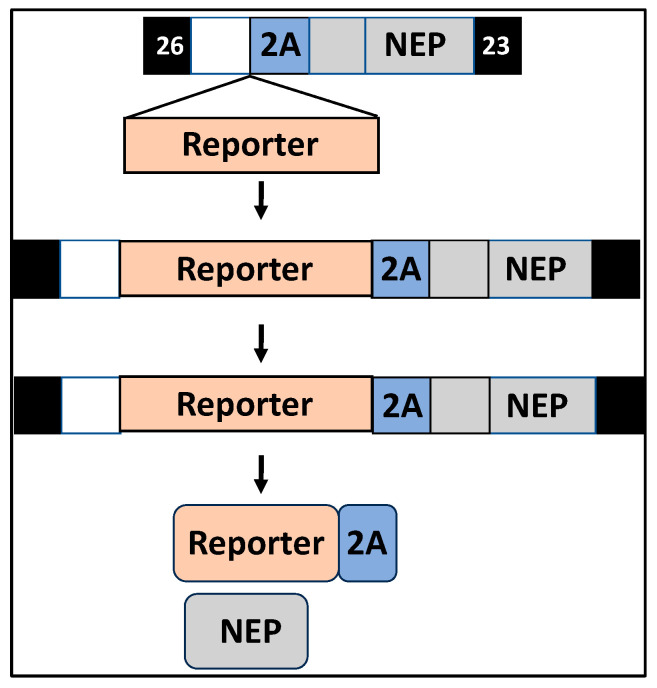
NS1-deficient rIAV expressing reporter genes: NS1-deficient rIAV expressing reporter genes are generated by substituting the NS1 ORF with the reporter gene (orange box) separated from the NEP ORF by the PTV-1 2A autoproteolytic cleavage site (blue box). The modified NS segment is transcribed as a single transcript and translated as a single polyprotein that is post-translationally processed at the PTV-1 2A site. The 2A peptide remains fused to the reporter gene. The 3′ and 5′ NCRs and packaging signals are indicated in black and gray boxes, respectively, as described in Figure 4.

**Table 1 ijms-25-10584-t001:** Replication-competent rIAV expressing reporter genes from the NS segment using the 2A autoproteolytic cleavage approach.

Gene	Virus Backbone ^(1)^	Transgene ^(2)^	Insertion Mechanism	Application	Reference
NS	pH1N1	Venus	2A site	Virus pathogenesis	[76]
NS	PR8	maxGFP	2A site	Virus pathogenesis	[77,78,79,80,81]
NS	PR8	maxGFP, turboRFP, Gluc	2A site	Antiviral and virus–host interaction	[10]
NS	PR8 pH1N1	mCherry	2A site	Antivirals, neutralizing antibodies, virus pathogenesis	[38]
NS	pH1N1	Timer	2A site	Virus propagation	[71]
NS	PR8 VN1203	Venus, eGFP, eCFP, mCherry	2A site	Virus–host interaction and virus pathogenesis	[56]
NS	PR8	Nluc, mCherry	ΔNS1 and 2A site	Virus–host interaction	[82]
NS	Guan2008	Nluc	2A site	Virus pathogenesis	[81]
NS	PR8 WSN	GFP, mCherry	2× 2A site	Virus pathogenesis and virus–host interaction	[44,45,57,73]

^(1)^ PR8: A/Puerto Rico/8/1934 (H1N1); pH1N1: A/California/04/2009 (H1N1); VN1203: A/Vietnam/1203/2004 (H5N1); Guan2008: A/Chicken/Guangdong/V/2008 H9N2; WSN: A/WSN/1933 H1N1. ^(2)^ GFP: Green fluorescent protein; maxGFP: advanced version of eGFP; Gluc: Gaussia luciferase; mCherry: monomeric Cherry fluorescent protein; Timer: modified Discosoma red fluorescent protein; Venus: advanced version of yellow fluorescent protein; eGFP: Enhanced GFP; eCFP: Enhanced cyan fluorescent protein; Nluc: NanoLuc luciferase.

**Table 2 ijms-25-10584-t002:** Replication-competent rIBV expressing reporter genes from the NS segment using the 2A autoproteolytic cleavage approach.

Gene	Virus Backbone ^(1)^	Transgene ^(2)^	Insertion Mechanism	Application	Reference
NS	B/Brisbane	maxGFP, mCherry, Timer	2A site	Antivirals, neutralizing antibodies, virus pathogenesis	[71,90]

^(1)^ B/Brisbane: B/Brisbane/60/2008 (Victoria lineage). ^(2)^ maxGFP: advanced version of eGFP; mCherry: monomeric Cherry fluorescent protein; Timer: modified Discosoma red fluorescent protein.

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
