# Peer review of "Recombinant Influenza A Viruses Expressing Reporter Genes from the Viral NS Segment"

_ijms, 2024, doi:10.3390/ijms251910584_

Round 1

Reviewer 1 Report

Comments and Suggestions for Authors

I do not have any additional comments.

Author Response

Reviewer #1:

Title: Recombinant influenza A viruses expressing reporter genes 2 from the viral NS segment

Reviewer Comments to Authors:  Studying influenza viruses from any aspects become very urgent, because of the increasing prevalence of various pandemic infections. Besides clinical aspects, viruses have relevance in basic science and gene therapy as reporters, expression vectors and gene carriers. In this review the authors summarize actual knowledge about viruses, as reporter vectors. Such a construct can be useful in many ways, for instance one can follow the presence, infection of the virus.

As a reviewer I have a couple of questions for the authors of this study.

  1. In the revised version, the authors should explain and give a schematic figure about the life cycle of the influenza viruses.

Response: We appreciate the suggestion made by the reviewer. Following the reviewer’s comment, we have added a schematic representation of the influenza virus life cycle as new Figure 2 in the new section 1.2.

  1. The authors should explain the difference in the life cycle of the Venus reporter virus compared to the natural one.

Response: We thank the reviewer for this suggestion. To our knowledge, both, natural influenza virus and the recombinant influenza virus expressing the venus reporter gene has the same life cycle, now illustrated in the new Figure 2 and described in the new section 1.2. We have clarified this point in the revised manuscript (section 2.1).

  1. Many virus reporter constructs are useful for drug screening or therapy. Is there any evidence that the mentioned constructs can be used for such purposes?

Response: Following the comment made by the reviewer, we have included information on how recombinant influenza viruses expressing reporter genes from the locus of the NS segment have been used in drug screenings.We have clarified this point in the revised manuscript (section 2.1).

  1. The authors agree that these vectors can be used in in vivo studies. Is there any study that shows their safety in such models?

Response: Following the comment made by the reviewer, we have included information on how recombinant influenza viruses expressing reporter genes from the locus of the NS segment have been used in in vivo studies. We have clarified this point in the revised manuscript (section 2.1).

  1. GFP is a well-known, general fluorescent protein used in molecular biology. What would be an advantage to use other stains such as the mentioned Venus or mCherry?

Response: We thank the reviewer for this comment. We have observed that recombinant viruses, including influenza virus, expressing GFP are less stable in maintaining the fluorescent protein than those expressing monomeric forms (e.g. Venus and mCherry). Venus is a monomeric form of the yellow fluorescent protein (YFP) and mCherry refers to monomeric Cherry fluorescent protein. So far, in our experience, GFP is not a good candidate to develop reporter-expressing recombinant IAV, at least when the NS segment is used to express the fluorescent protein.

I think this paper is excellent and is an important addition to the literature about reverse genetics. The article is well written, it explores and analyzes a topic that has relevance in today's science.

Response: We thank the reviewer for the final comment, including the relevance to the literature and to the field.

Reviewer 2 Report

Comments and Suggestions for Authors

In manuscript entitled “Recombinant influenza A viruses expressing reporter genes from the viral NS segment”, Luis Martinez-Sobrido and Aitor Nogales summarized recombinant IAV expressing reporter proteins using NS gene. I agree that this type of manuscript is useful as a reference but if the author could summarize all the example of recombinant IAV expressing reporter genes, which would be much more valuable. I don’t see the advantage to stick only to NS segment recombinants. Specific comments follow.

Major points:

1.     Line 13: Please describe how easy to detect rIAV in vitro and in vivo by number of lower limits of detection particles and duration after infection.

2.     Lines 193-196: I don’t understand why the authors say “For in vivo studies, luciferase proteins represent a better option to detect the presence of the virus in the entire infected animals using IVIS” by saying “However, they required the administration of luciferase substrates.”

Minor points:

1.     Line 144 & Figure 2: “PA” is already used for “Polymerase Acidic (PA)”.

Author Response

Reviewer #2:

General comment: In manuscript entitled “Recombinant influenza A viruses expressing reporter genes from the viral NS segment”, Luis Martinez-Sobrido and Aitor Nogales summarized recombinant IAV expressing reporter proteins using NS gene. I agree that this type of manuscript is useful as a reference but if the author could summarize all the example of recombinant IAV expressing reporter genes, which would be much more valuable. I don’t see the advantage to stick only to NS segment recombinants. Specific comments follow.

Response: We thank the reviewer for the overall comment that our manuscript is useful. We concur with the reviewer that our manuscript would be more valuable if we summarize all the examples of recombinant IAV expressing reporter genes. However, the overall goal of this review article is to describe those recombinant IAV expressing reporter genes from the locus of the viral NS segment. Describing all the recombinant IAV expressing reporter genes described in the literature, including different viral types, subtypes, and strains; different reporter genes; and experimental strategies for the expression of the reporter genes from different viral segments, will lack focus. Thus, we believe that our review article describing recombinant IAV expressing reporter genes from the viral NS segment provides some focus to the review. We have indicated in our manuscript that recombinant IAV expressing other reporter genes using similar or alternative strategies from different viral segments have been described in the literature and we have included some relevant references.

Major points:

  1. Line 13: Please describe how easy to detect rIAV in vitro and in vivo by number of lower limits of detection particles and duration after infection.

Response: We thank the reviewer for this constructive suggestion. Following the recommendation made by the reviewer, we have included this information in the revised manuscript. However, due to word limitations in the abstract, we have clarified this point in the new section 2.1 of the revised manuscript.

  1. Lines 193-196: I don’t understand why the authors say “For in vivo studies, luciferase proteins represent a better option to detect the presence of the virus in the entire infected animals using IVIS” by saying “However, they required the administration of luciferase substrates.”

Response: We apologize for the lack of clarity. We and others have previously shown that recombinant viruses, including influenza virus, expressing luciferase proteins can be used to track viral infection in vivo, in the entire animals, using in vivo imaging systems, contrary to the situation with recombinant viruses expressing fluorescent proteins. However, a limitation of using luciferase-expressing recombinant viruses to track viral infection in vivo is the need of administrating the luciferase substrate, contrary to the situation with ex vivo imaging of recombinant viruses expressing fluorescent proteins that do not require the administration of a substrate. We have clarified this point in the section 2.1 of the revised manuscript.

Minor points:

  1. Line 144 & Figure 2: “PA” is already used for “Polymerase Acidic (PA)”.

Response: We thank the reviewer and apologize for this editorial mistake that has been corrected in the revised manuscript.

Round 2

Reviewer 2 Report

Comments and Suggestions for Authors

No further comments to the authors, all ques􀆟ons and comments were answered to my satisfaction.